# Nobiletin Mitigates D-Galactose-Induced Memory Impairment via Improving Hippocampal Neurogenesis in Mice

**DOI:** 10.3390/nu15092228

**Published:** 2023-05-08

**Authors:** Wei Xiong, Rongzi Li, Boying Li, Xifan Wang, Huihui Wang, Yanan Sun, Xiaoyu Wang, Yixuan Li, Fazheng Ren

**Affiliations:** 1Key Laboratory of Precision Nutrition and Food Quality, Department of Nutrition and Health, China Agricultural University, Beijing 100083, China; xiongwei910702@126.com (W.X.); rzl0903@126.com (R.L.); lboying@126.com (B.L.); wangxfan@126.com (X.W.); whhlyh1227@163.com (H.W.); 15153515695@163.com (Y.S.); liyixuan@cau.edu.cn (Y.L.); 2Food Laboratory of Zhongyuan, Luohe 462000, China; 3Key Laboratory of Functional Dairy, Co-Constructed by Ministry of Education and Beijing Municipality, College of Food Science & Nutritional Engineering, China Agricultural University, Beijing 100083, China; xy.wang@cau.edu.cn

**Keywords:** aging, nobiletin, memory, neurogenesis, neuroinflammation

## Abstract

Memory impairment is a characteristic of brain aging, and it is associated with a decrease in neurogenesis. Therefore, enhancing neurogenesis is a potential method for mitigating brain aging. Nobiletin (NOB) is a natural polymethoxylated flavonoid derived from citrus peels. It acts as an antioxidant, enhances anti-inflammation, and displays neuroprotective properties. However, the mechanism of NOB on brain aging has not been elucidated. In this study, D-galactose-induced aging mice were treated with NOB (100 mg/kg/day) for 10 weeks. NOB administration attenuated D-galactose-induced memory impairment and restored hippocampal neurogenesis, including the number of newborn neurons and neural stem cells in mice. Furthermore, it downregulated the pro-inflammatory mediators IL-1 β, IL-6, and pP65 (by 42.2%, 22.9%, and 46.4% of those in the D-galactose treated group, respectively) in the hippocampus and blocked microglia and astrocyte activation. In vitro, NOB inhibited D-galactose-induced inflammatory responses in BV2 cells, and the conditioned medium prepared from NOB- and D-galactose-co-treated BV2 cells elevated the viability (90.3% of control) and differential ability (94.9% of control) of C17.2 cells, compared to the D-galactose-treated group alone. It was concluded that NOB could restore memory impairment via the improvement of neurogenesis by ameliorating neuroinflammation in the hippocampus. Overall, NOB is a potential candidate neurogenesis enhancer for improving brain function.

## 1. Introduction

Aging is a process characterized by the accumulation of functional impairment at the cellular and organism levels. According to the World Health Organization report (the 2022 revision), the global elderly population is growing faster than all other groups. The population of people aged 60 years and older all over the world will have increased to 2.1 billion by 2050 [1]. Although many people are excited about the potential of a longer lifespan, this must be accompanied by a longer period of health rather than additional years of infirmity and sickness.

Indeed, neurodegeneration can easily appear during aging, which is associated with enormous personal and socioeconomic costs [2]. The combined disruption of homeostatic processes, such as mitochondrial dysfunction, protein aggregation, genomic instability, and lysosomal dysfunction, contributes to brain aging [3]. In recent years, an attractive theory has been proposed, which states that the age-related declines in brain function are related to decreased neurogenesis [4]. The process of neurogenesis creates new neurons in the brain, which involves neural stem cells’ (NSCs) proliferation and differentiation. The subventricular zone (SVZ) and the dentate gyrus (DG) of the hippocampus are the two well-studied neurogenic niches. During the aging process, the number of NSCs decreases, leading to a reduction in neurogenesis [5]. It is speculated that the decrease in hippocampal neurogenesis might be an important inducement of cognitive impairment in the aged subjects, including novel object recognition, associative learning, and even the establishment of long-term memory [6,7]. Therefore, improving neurogenesis in the elderly brain might be a potentially effective therapeutic approach to resist or mitigate age-related cognitive decline.

The decline in neurogenesis, which occurs during the normal aging process, may be attributed to a concomitant increase in neuroinflammation [8]. As previously mentioned, adult NSCs reside in specialized microenvironments referred to as stem cell niches, which are important for the maintenance of neurogenesis. Hippocampal neuroinflammation, via the release of a multitude of inflammatory molecules, is increasingly recognized as a critical pathophysiological factor that has a negative impact on neurogenic niche dynamics [8]. Disturbed neurogenic niches affect the normal generation and differentiation of new neurons and compromise the survival of hippocampal neural stem cells.

The protective effects of herbal extracts in aging-induced cognitive impairments have been extensively studied. Nobiletin (NOB), a natural polymethoxylated flavonoid extracted from the fruit peel of citrus, has shown efficacy in antioxidant, anti-inflammatory, antibacterial, and anti-diabetic properties [9,10]. Previous studies have uncovered that NOB exhibits neuroprotective properties. In lipopolysaccharide-treated BV-2 microglia cells, NOB intervention reduced the level of pro-inflammatory cytokines and suppressed excessive microglial activation [11]. In animal models, NOB ameliorated Aβ-induced cognitive deterioration by inhibiting neuroinflammation and oxidative stress [12]. Moreover, in human neural progenitor cells (hNPCs), NOB treatment restored sodium-arsenate-induced neurite damage and altered neuronal marker expression, suggesting that NOB possesses a protective and restorative potential in hNPCs [13].

The available evidence suggests that NOB has multifaceted physiological activity and exerts neuroprotective effects on the aging brain. However, the underlying molecular mechanisms of the neuroprotective effects of NOB remain mostly elusive. This study aimed to elucidate the underlying mechanisms of NOB to protect against D-galactose-induced brain aging, providing evidence to support the therapeutic potential of NOB for the aging brain. A subcutaneous injection of D-galactose can induce aging and is widely used in anti-aging intervention studies [14]. The accumulation of D-galactose can elevate the generation of reactive oxygen species (ROS), leading to oxidative stress, inflammation, mitochondrial dysfunction, and eventually the aging of an organism [15]. Moreover, D-galactose-induced brain aging is similar in many ways to natural brain aging. It has been reported that mice given prolonged injections of D-galactose for 6–10 weeks show progressive declines with respect to learning and memory; an increase in neuroinflammation and oxidative stress; and neurogenesis reduction and neuronal degeneration [15]. In this study, we administered D-galactose in mice once a day for 10 weeks, artificially inducing aging in mice.

## 2. Materials and Methods

### 2.1. Animals

C57BL/6J mice (male, 8-week-old) were purchased from Beijing HFK Bioscience Co. Ltd. (Beijing, China). They were kept in typical laboratory settings with a 12-h light/dark cycle and a temperature of 24–26 °C. Water and food were freely available to the mice. After one week of acclimation, mice were randomly assigned into four groups (*n* = 15): (1) control, (2) control+NOB, (3) D-gal, and (4) D-gal+NOB. Subcutaneously injected D-galactose (500 mg/kg/day, G5388, Sigma, St. Louis, MI, USA) was dissolved in a 0.9% saline solution. NOB (100 mg/kg/day, IN0210, MedChemExpress, Monmouth Junction, NJ, USA) was dissolved in DMSO, then in 0.9% saline. NOB was taken by oral gavage following D-galactose injection. The solutions were administered at a volume of 8 mL/kg body weight. Specifically, the control group received a subcutaneous injection of 0.9% saline and an oral gavage of 0.9% saline (containing 9.6% DMSO); mice in the control+NOB group received a subcutaneous injection of 0.9% saline and an oral gavage of NOB daily; mice in the D-gal group received a subcutaneous injection of D-galactose and an oral gavage of 0.9% saline (containing 9.6% DMSO); and mice in the D-gal+NOB group received a subcutaneous injection of D-galactose and an oral gavage of NOB (Figure 1).

### 2.2. Behavioral Testing

Novel object recognition (NOR) tests and Y-maze spontaneous alternation tests were used to examine memory function. We carried out NOR tasks in accordance with previous studies [16]. The basis of these tasks is the spontaneous tendency of mice to prefer to discover a novel object than a familiar one. Both objects have the same height and volume, but their shape and appearance vary. In this study, 24 h before the NOR test, the animals freely roamed around in an empty arena as they became accustomed to it. The formal tests included familiarization and choice trials. Mice were placed in the arena to explore two identical objects (1 and 2) for 5 min during the familiarization trial, while mice were allowed to explore a familiar object and a novel object for 5 min during the choice trial. There was a 1-h interval time between the two trials. In all studies, the animals’ item exploration duration, which was defined as touching or sniffing the objects within 2 cm, was measured.

The Y-maze was performed according to a previous study [17]. Testing occurred in a Y-shaped maze with three identical arms. For assessment of working memory, the animals were placed in the center of the maze with all arms open, and were allowed to freely explore all arms for 7 min. We recorded the number of mice entries, and the sequence of arms entered. A spontaneous alternation occurs when mice sequentially enter each of the three arms (e.g., ABC, BCA, CAB, etc.).

### 2.3. Tissue Collection

The animals were anesthetized by CO_2_, then perfused with 0.9% saline solution. Brains from 8 mice of each group were rapidly dissected, and the hippocampus collected, and then stored at −80 °C for future Western blot analyses. Brains from 7 mice of each group prepared for later immunofluorescence analysis were collected and fixed in a 4% paraformaldehyde for 48 h.

### 2.4. Immunofluorescence

The fixed brain tissues were embedded in paraffin. Then, tissues underwent a coronal section (5 µm) at the hippocampal level according to the Mouse Brain Atlas [18]. Sections were then placed on slides. For each group of mice, similar brain sections (≥3 per mouse) containing the hippocampus region were chosen. Deparaffinized brain sections were rehydrated in graded alcohol solutions after being deparaffinized in xylene. After rinsing in phosphate-buffered saline (PBS, pH 7.4), the sections were incubated with primary antibodies overnight at 4 °C. On the next day, after washing with 0.01% TritionX-100, brain sections were reacted with secondary antibody for 1 h. Finally, the slides were stained with DAPI and cover-slipped. The primary antibodies rabbit anti-SOX2 (1:100, 11064-1-AP) and mouse anti-Iba1 (1:100, 10904-1-AP) were provided by Proteintech (San Diego, CA, USA). The rabbit anti-DCX antibody (1:100, A0149) was purchased from ABclonal (Woburn, MA, USA), and the mouse anti-GFAP (1:150, ab7260) was purchased from Abcam (Cambridge, UK). In addition, the secondary antibodies Alexa Fluor 647-labeled goat anti-mouse and anti-rabbit IgG (1:500, A0473 and A0468) were purchased from Beyotime (Shanghai, China).

The brain slides were visualized under an LSM900 with the Airyscan confocal microscope (ZEISS), which has an AxioCam 702 mono camera. Magnifications of 10× and 20× were applied. Images were captured by Zen Blue 3.1 software and then analyzed via Image J software (version 1.53t). We selected the DG zone of the hippocampus for imaging, and three visual fields for each section were taken. The cell number/intensity was determined at the same area under magnification and ≥3 brain sections per mouse were chosen in order to offer normalized data in the DG zone. We identified NSCs as SOX2+ nucleus cells in the DG zone. The DCX antibody marks the cytoplasm, and DCX+ cells located in the subgranular layer of the DG were identified as newborn neurons. GFAP and Iba1 fluorescence intensities in the DG region were measured by Image J software.

### 2.5. Western Blot Analysis

The tissues were homogenized in a RIPA (P0013C) lysis buffer containing 1 mM PMSF (ST507). A BCA kit (P0012) was used to measure the concentration of the total protein. In this study, 20–30 ug of proteins from each sample was loaded onto a sodium dodecyl sulfate (SDS) gel for electrophoresis and then transferred to polyvinylidene difluoride (PVDF, 03010040001, Sigma, St. Louis, MI, USA) membranes. Subsequently, the blots were incubated with the primary antibodies overnight at 4 °C. On the second day, the membrane was washed and then reacted with the secondary antibody for 1 h at room temperature. Finally, the blots were visualized with an enhanced luminol-based chemiluminescent kit (34075, Thermo Fisher, Waltham, MA, USA) and scanned by an electronic camera system. The primary antibodies rabbit anti-β-actin (1:1000, 4970S), rabbit anti-IL-6 (1:1000, 12912T), rabbit anti-IL-1β (1:1000, 12703S), and rabbit anti-pP65 (1:1000, 3033S) were purchased from Cell Signaling Technology (Danvers, MA, USA). The primary rabbit antibody P65 (1:1000, Ab16502) was purchased from Abcam (Cambridge, UK). The secondary antibodies goat anti-rabbit IgG (H+L) HRP (1:5000, A0218) and goat anti-mouse IgG (H+L) HRP (1:5000, A0216), RIPA, PMSF, and the BCA kit were purchased from Beyotime (Shanghai, China).

### 2.6. Preparation of Conditioned Medium

For routine cultures, BV2-immortalized murine microglia cells (Shanghai Cell Research Center, Shanghai, China) were maintained in Dulbecco’s modified Eagle’s medium (DMEM, 41965039, Gibco, Grand Island, NE, USA) supplemented with 10% fetal bovine serum (FBS, 12664025C, Gibco, Grand Island, NE, USA) and 100 U/mL penicillin–streptomycin (C0222, Beyotime, Shanghai, China) at 37 °C in a 5% humidified CO_2_ incubator. To obtain BV2 conditioned medium (CM), 1 × 10^5^ cells/mL of cells were cultured in several 12-well plates with DMEM at 2% FBS. D-galactose alone or together with NOB was applied to the culture medium (the concentration of D-galactose was determined by a cell viability assay). The CM was collected after 24 h and stored at −20 °C. The CM was thawed at 4 °C before use.

### 2.7. Determine the Level of Inflammatory Cytokines in CM

The levels of IL-4 (IL-4, RXSWSXB203051M) and IL-10 (IL-10, RX-G203075M) in the CM of BV2 were determined by Enzyme-Linked Immunosorbent Assay (ELISA) kits. The kits were purchased from Ruxin Biotech (Quangzhou, China). ELISA was carried out according to the manufacturer’s instructions.

### 2.8. C17.2 Cell Culture in the Presence of CM

For routine cultures, the C17.2 cells (Shanghai Cell Research Center, Shanghai, China) were cultured in DMEM supplemented with 10% FBS and 100 U/mL penicillin–streptomycin. For differentiation studies, the cells were seeded in a routine culture medium. A total of 24 h after seeding, the medium was changed to the BV2-derived CM and supplemented with 10 ng/mL NGF (N0513, Sigma, St. Louis, MI, USA) and 10 ng/mL BDNF (B-250, Alomone labs, Jerusalem, Israel). Every third day, the differentiation medium was changed. C17.2 cell viability was determined using a cell viability assay. The differential ability of C17.2 cells was observed under microscopy, and neurites’ length was quantified by the tracing method (Simple Neurite Tracer, Image J).

### 2.9. Cell Viability Assay

BV2 cells were seeded into a 96-well plate for 12 h and then treated with D-galactose at various concentrations (100 ug/mL, 200 ug/mL, 1 mg/mL, 5 mg/mL, 10 mg/mL, 25 mg/mL, and 50 mg/mL) for 24 h. C17.2 cells were also seeded into a 96-well plate for 12 h and then treated with BV2-derived CM. Cell viability was detected via a cell-counting kit-8 (CCK-8) assay (C0037, Beyotime, Shanghai, China). Briefly, 10 uL of the CCK-8 reagent was added to each well and cultured for 1 h at 37 °C. The absorbance was measured at 450 nm using a microplate reader (Bio-Rad, Hercules, CA, USA).

### 2.10. Quantitative PCR (qPCR) Analysis

The BV2 cells’ RNA was extracted for qPCR for the detection of IL-6, IL-1β, and TNF-α. The total RNA was purified using the Trizol reagent (Invitrogen, Carlsbad, CA, USA). RNA was reverse transcribed to cDNA by 5× All-In-One RT Master Mix (G592, Applied Biological Materials, Richmond, BC, Canada). Next, qPCR reactions were conducted using Step One Plus (Applied Biosystems, Waltham, MA, USA) with the Takara TB Green Fast qPCT mix (RR430, Takara, Shiga, Japan). The qPCR primer sequences are listed:
*TNF-α*:Fwd-5′GGAACACGTCGTGGGATAATG3′
Rev-5′GGCAGACTTTGGATGCTTCTT3′*IL-6*:Fwd-5′GCAGCATCACCTTCGCTTAGA3′
Rev-5′CAGATATTGGCATGGGAGCAAG3′*IL-1β*:Fwd-5′GCAACTGTTCCTGAACTCAACT3′
Rev-5′ATCTTTTGGGGTCCGTCAACT3′*Gapdh*:Fwd-5′TGGATTTGGACGCATTGGTC3′
Rev-5′ TTTGCACTGGTACGTGTTGAT3′

### 2.11. Statistical Analysis

All data are presented as the mean ± standard error of the mean (SEM) of at least three independent experiments performed on different days, and *p* < 0.05 was considered statistically significant. For the NOR test analysis, a Student’s *t* test (two-tailed unpaired) comparing both familiar and novel object for each group independently was carried out. For other studies, we performed post hoc testing using Tukey’s HSD after one-way ANOVA (GraphPad Prism, San Diego, CA, USA). Variables in different groups are identified by one or more letters. Variables are statistically significant if they have different letter, while variables are statistically indistinguishable if they share at least one letter.

## 3. Results

### 3.1. Effect of NOB on Hippocampal-Dependent Memory Function in D-Gal-Treated Mice

The NOR task was used to evaluate mice recognition memory. In this study, results from the familiarization trial suggest that animals from the four groups spent similar times exploring object one and object two (*p* > 0.05, Figure 2A). Furthermore, in the choice trial, mice in the control and control+NOB groups spent a significantly longer time exploring the novel object than the familiar object (*p* < 0.05, Figure 2B). In contrast, mice in the D-gal group showed a similar preference for both the familiar and novel objects (*p* > 0.05, Figure 2B), suggesting that D-galactose induced impaired recognition memory. However, after NOB administration, D-galactose-treated mice preferred novel objects in the choice trial (*p* = 0.03, Figure 2B), which indicates that NOB ameliorated D-galactose-induced recognition memory deterioration.

To determine the working memory of mice, the spontaneous alternation behavior of animals was tested by Y-maze (Figure 3A). Mice in the D-gal and D-gal+NOB groups had a smaller number of arm entries compared to the control group (Figure 3B). The analysis of spontaneous alternations confirmed an impairment in working memory in D-galactose-injected mice (*p =* 0.009, compared to the control; Figure 3C). However, the spontaneous alternation ratio was elevated by 80.9% after the NOB treatment (*p* = 0.04, Figure 3C). Hence, similarly to recognition memory, working memory performance declined in mice injected with D-galactose, and NOB administration was sufficient to protect against these deficits in mice.

### 3.2. Hippocampal Neurogenesis in D-Galactose or/and NOB-Treated Mice

#### 3.2.1. Effect of NOB on New Neuron Generation in the Hippocampus of D-Gal-Treated Mice

The decrease in hippocampal neurogenesis was suggested to be associated with impaired memory performance [7]. To verify the influence of D-galactose on adult hippocampal neurogenesis and whether NOB can promote neurogenesis, immature neurons in the subgranular zone of the DG in the hippocampus were quantified by DCX-positive (DCX+) cells. The staining showed that a 65.9% reduction (*p* = 0.003, Figure 4) in DCX+ cells was detected in the D-gal group compared with the control group, whereas DCX+ cells in the D-gal+NOB group were restored to 72% of the control group (Figure 4). These outcomes indicate that NOB improves neurogenesis in the hippocampus of D-galactose-induced aging mice.

#### 3.2.2. Effect of NOB on Neural Stem Cell Number in the Hippocampus of Mice Treated with D-Galactose

Next, we investigated the SOX2-positive (SOX2+) neural stem cell population throughout the DG zone of the hippocampus. Our results confirmed that the number of SOX2+ cells in mice receiving D-galactose alone was reduced by 35.5% compared to the control group (*p* = 0.001). Nevertheless, mice that received both D-galactose and NOB expressed 31.4% higher SOX2+ cell numbers than the D-gal group (*p* = 0.007, Figure 5). These results revealed that co-administration with NOB could protect against the D-galactose-induced decrease in neural stem cells in the mouse hippocampus.

### 3.3. Effect of NOB on Neuroinflammation in the Hippocampus of D-Gal-Treated Mice

To identify the role of NOB in D-galactose-induced hippocampal inflammation, we assessed the levels of the inflammatory mediators IL-6, IL-1 β and pP65 in mice by Western blotting. D-galactose-treated mice had 48.6% (*p* = 0.001), 32.4% (*p* = 0.002), and 51.7% (*p* = 0.001) higher hippocampal levels of the pro-inflammatory mediators IL-1β, IL-6, and pP65/P65, respectively, compared with the control group (Figure 6). Nevertheless, NOB was able to counteract D-galactose-induced IL-1β, IL-6, and pP65 elevation, resulting in a 42.2% (*p* = 0.002), 22.9% (*p* = 0.004), and 46.4% (*p* = 0.003) reduction, respectively (Figure 6). Moreover, we examined the intensities of Iba-1 and GFAP proteins and their correlation to the levels of activated microglia and astrocytes, respectively. We observed that D-galactose significantly enhanced the level of activated astrocytes and activated microglia. However, both astrocyte and microglia reactivities were inhibited after the local administration of NOB, suggesting that NOB is an important neuroinflammation modulator (Figure 7).

### 3.4. Effect of NOB on Inflammatory Responses in BV2 Cells Induced by D-Galactose

In order to clarify the effects of activated microglia on neural stem cells, we used BV2 cells and C17.2 cells to represent microglia and neural stem cells, respectively (Figure 8A). BV2 cells were treated with D-galactose to mimic neuroinflammation in the hippocampus, with NOB supplementation as an intervention. Firstly, we cultured BV2 cells in the presence or absence of varying doses of D-galactose and examined the cell viability after 24 h of administration. Exposure to D-galactose significantly decreased cell viability in a dose-dependent manner from upwards of 10 mg/mL of D-galactose (Figure 8B). A higher dose of D-galactose (25 mg/mL) resulted in a considerable cell death, as observed under microscopy. D-galactose at a concentration of 10 mg/mL was selected in subsequent studies. Next, in order to confirm that the activation of BV2 cells triggered the release of pro-inflammatory cytokines, BV2 cells were collected, and qPCR was used to evaluate the mRNA expression of pro-inflammatory cytokines. In addition, the culture medium from 24-h D-galactose-treated BV2 cells was collected. An ELISA was conducted to determine the release of IL-10 and IL-4 from activated microglia. As expected, treatment with D-galactose significantly increased the expression levels of IL-6 (79.2%, *p* = 0.003, Figure 8C), IL-1β (46.6%, *p* = 0.001, Figure 8D), and TNF-α (40.1%, *p* = 0.004, Figure 8E) in BV2 cells and decreased the release of IL-10 (35.1%, *p* = 0.003, Figure 8F) and IL-4 (35.6%, *p* = 0.02, Figure 8G) by BV2 cells compared to the control group. However, co-treatment with NOB protected BV2 cells from D-galactose damage and reduced the inflammation response. In addition, we incubated C17.2 cells with microglia-derived CM to investigate whether the alleviation of microglial inflammation by NOB contributes to the survival of neural stem cells. The CM derived from D-galactose-treated BV2 cells, with or without NOB treatment, was added to C17.2 cells. D-galactose-induced CM from BV2 cells resulted in a 35% (*p* = 0.007, Figure 9A) decrease in C17.2 cell viability. However, NOB supplementation effectively restored the cell viability of the C17.2 cells to 90.3% of the control (Figure 9A). In addition, different CMs supplemented with NGF and BDNF were used to differentiate C17.2 cells. Our results demonstrated that C17.2 cells cultured in D-galactose-induced CM showed a significantly lower differential ability, as evidenced by the reduced average neurite length (44.4%, *p* = 0.002, Figure 9B,C), compared with the control group. In contrast, CM derived from the D-gal+NOB group improved C17.2 cell differentiation (90.6%, *p* = 0.002, Figure 9B,C) compared with the D-gal group.

## 4. Discussion

Declines in cognition are commonly experienced with aging. NOB, a major flavonoid polymethoxylated flavone found in citrus peels, shows protective roles in Alzheimer’s disease, Parkinson’s disease, and aging mice models [19,20]. Our results showed that the NOB treatment could alleviate memory deficits induced by D-galactose in mice, which is consistent with previous studies that NOB could ameliorate memory deficiency in aging animals [20,21]. Next, we investigated the underlying mechanism for the regulation of the aging brain by NOB, and our results showed that NOB acts as a neurogenesis enhancer to alleviate cognitive deficits.

Recognition memory and spatial memory depend on the function of the hippocampus [22]. NOR tests can assess hippocampus-dependent recognition memory, which refers to the ability to identify previously encountered objects or events [23]. Recent studies have indicated that mice treated with D-galactose have a poor performance in NOR tests [24]. This finding is consistent with our study that D-gal-group mice displayed impaired recognition behavior, as evidenced by the observed insignificant preference for the novel object and the familiar object. In contrast, mice administered with D-galactose+NOB preferred the novel object more than the familiar one. The Y-maze test is employed to investigate spatial memory in mice. By allowing mice to freely explore all three arms of the maze, spontaneous alternations are recorded in order to evaluate spatial memory [23]. Mice with good spatial memory will recall the arms that they have already come across. The hippocampus is involved in this memory process. Similar to NOR tests, this study is consistent with previous research demonstrating that D-galactose leads to decreased spontaneous alternations in mice [25]. NOB supplementation enhanced the spontaneous alternation behavior, indicating that NOB restored the spatial memory of the D-galactose-injected mice.

NSCs in the DG of the hippocampus can proliferate and generate intermediate progenitor cells, which can produce neuroblasts that subsequently differentiate into mature neurons [7]. Newborn neurons exhibit extraordinary plasticity as they mature, and they respond to environmental cues via complex molecular regulatory networks. Despite their low numbers, adult-born neurons have an essential impact on hippocampal activities because of their higher excitability [26], and they have been found to improve learning and memory abilities in rodents. It is well documented that adult neurogenesis decreases during physiological aging [27], and it contributes to cognitive declines during aging [28]. Recent studies have suggested that NOB may have the potential to improve neurogenesis. In NaAsO2-induced hNPCs, NOB treatment restored neurite damage, the level of stress granule markers, and impaired neurogenesis [13]. Hesperidin, another flavonoid found in citrus peels, has been shown to improve hippocampal neurogenesis and memory function in valproic-acid-injected mice [27]. In our artificial aging model, the level of neurogenesis was significantly lower than in control mice. However, NOB administration reversed the reduction in newborn neurons and neural stem cells. Therefore, our results suggest that NOB could ameliorate hippocampal neurogenesis disruption during aging.

Changes to the physiology of the body accompany aging, including peripheral and central nervous system (CNS) inflammation, which can negatively impact neurogenesis [29]. The hippocampus is sensitive to the potentiated neuroinflammatory responses [29]. In the DG zone of the hippocampus, there is a specialized neurogenic niche defined as a microenvironment in which neurogenesis occurs. Microglia, astrocytes, neuroblasts, granule cells, and NSCs are key components of this niche [6]. Cells in the neurogenic niche respond to extrinsic cues and secrete various factors to regulate the entire neurogenesis process, including the activation and proliferation of NSCs, as well as the generation and integration of new neurons into existing neural networks. During aging, the inflammation level increases in the neurogenic niche, and it is characterized by T-cell infiltration, elevated inflammatory cytokines levels, and activated microglia and astrocytes [30]. In normal conditions, astrocytes make direct contact with NSCs and release signaling molecules to modulate NSCs’ fate. However, in aging or pathological situations, astrocytes undergo molecular and functional remodeling and become activated, upregulating the production of pro-inflammatory cytokines and decreasing neuronal fitness [31]. Microglia are the main resident immune surveillants in the CNS, and they are responsible for neuron cell development and maintenance [32]. In response to aging-induced chronic inflammation, activated microglia exaggerate the release of pro-inflammatory cytokines, such as TNF-α and IL-1 β [32]. All of these play detrimental roles in the maintenance of hippocampal neurogenesis [17]. In our study, we detected an increased level of the pro-inflammatory cytokines IL-6 and IL-1β in the hippocampus of D-galactose-induced mice. However, NOB administration restored this alteration in the hippocampus. In addition, we found that D-galactose could result in the excessive activation of microglia and astrocytes, as evidenced by the elevated intensity of Iba-1 and GFAP. Consistent with the results of pro-inflammatory cytokines, NOB decreased the intensity of Iba-1 and GFAP.

Furthermore, our in vitro study indicated that the levels of the pro-inflammatory cytokines IL-1β, IL-6, and TNF-α were significantly upregulated in BV2 cells. Additionally, the secretion of the anti-inflammatory cytokines IL-10 and IL-4 was significantly downregulated in the culture medium of D-galactose-treated BV2 cells. Then, we observed that C17.2 cells exposed to a D-galactose-treated BV2-cell CM displayed decreased cell viability and differential ability. Therefore, we speculated that the inflammatory response in D-galactose-induced CM may contribute to C17.2 cell injury and inhibit their differentiation. However, NOB-co-treated CM restored C17.2 cells’ viability and differentiation rate, suggesting the neuroprotective effect of NOB. Overall, our results suggest that NOB ameliorates D-galactose-induced hippocampal neuroinflammation and improves the homeostasis of neurogenic inches; this may be responsible for the enhanced neurogenesis in the D-gal+NOB group mice.

Although hippocampal-dependent memory is closely related to hippocampal neurogenesis, other interdependent factors work together leading to brain aging and impaired cognitive function [33]. For example, mitochondrial dysfunction and excessive oxidative stress, as well as impaired DNA repair, are key hallmarks of brain aging [33]. Considering the complexity of aging, interventions focusing on one target may not be enough. Therefore, it is also necessary to explore the effect of NOB on other hallmarks of brain aging.

## 5. Conclusions

In summary, our study confirmed that D-galactose reduces hippocampal neurogenesis and leads to memory impairment in adult mice. Nevertheless, NOB could regress these deteriorations in D-galactose-induced aging mice through improving neurogenesis by ameliorating neuroinflammation in the hippocampus. Overall, NOB is a potential candidate neurogenesis enhancer for preventing brain aging.

## Figures and Tables

**Figure 1 nutrients-15-02228-f001:**
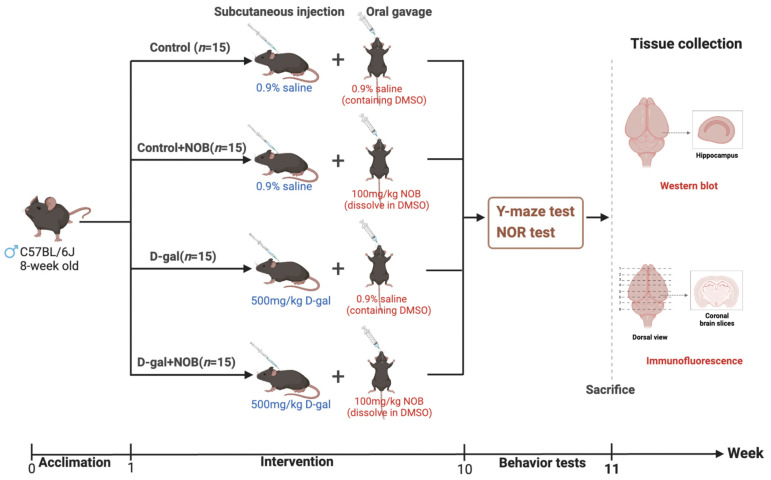
Schematic illustration of animal experimental design.

**Figure 2 nutrients-15-02228-f002:**
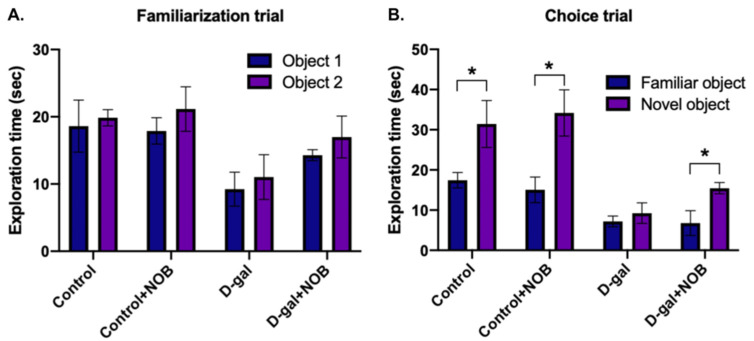
NOB ameliorated the recognition memory deficits caused by D-galactose. (**A**) Objects’ exploration time in the familiarization trial. (**B**) Objects’ exploration time in the choice trial. Data are shown as the mean ± SEM (*n* = 6). Student’s *t* test. *, *p* < 0.05.

**Figure 3 nutrients-15-02228-f003:**
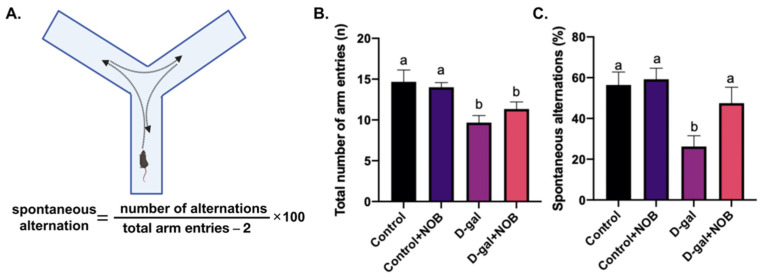
NOB ameliorated the spatial memory deficits caused by D-galactose. (**A**) Diagram of the Y-maze test. (**B**) The total number of arm entries in the four groups. (**C**) The proportion of spontaneous alternations in the four groups. Data are shown as the mean ± SEM (*n* = 6), from a one-way ANOVA with Tukey’s post hoc test. Different lowercase letters (a, b) represent significant differences between groups (*p* < 0.05).

**Figure 4 nutrients-15-02228-f004:**
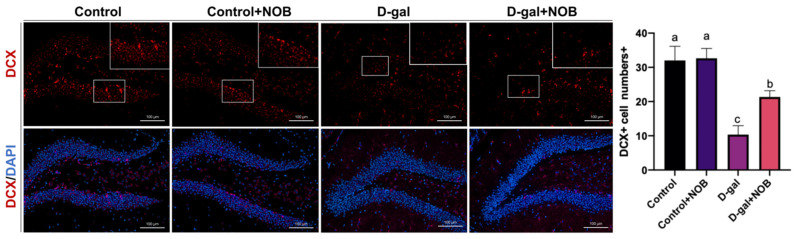
NOB elevated the number of newborn neurons in D-galactose-treated mice. Representative images of DCX-positive cells in the dentate gyrus of hippocampus and the corresponding quantification graph. Values are shown as the mean ± SEM (*n* = 4), from a one-way ANOVA with Tukey’s post hoc test. Different lowercase letters (a, b, c) represent significant differences between groups (*p* < 0.05).

**Figure 5 nutrients-15-02228-f005:**
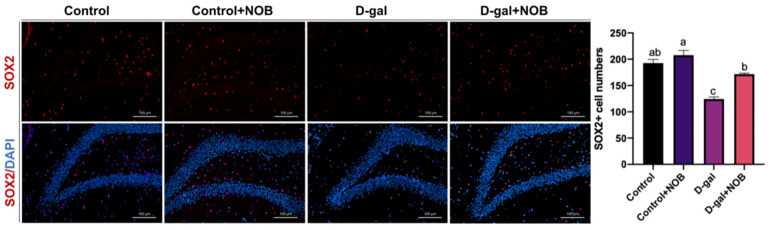
NOB elevated the number of neural stem cells in D-galactose-treated mice. Representative images of SOX2-positive neural stem cell population in the dentate gyrus of hippocampus and the corresponding quantification graph. Data are shown as the mean ± SEM (*n* = 4), from a one-way ANOVA with Tukey’s post hoc test. Different lowercase letters (a, b, c) represent significant differences between groups (*p* < 0.05).

**Figure 6 nutrients-15-02228-f006:**
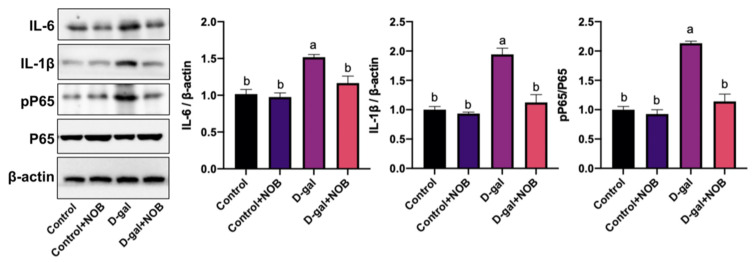
NOB ameliorated the levels of pro-inflammatory mediators in the hippocampus of D-galactose-treated mice. Representative images of Western blot analysis of IL-1β, IL-6, and pP65 in the hippocampus of the four groups of mice and corresponding quantification graphs. Values are shown as the mean ± SEM (*n* = 4), from a one-way ANOVA with Tukey’s post hoc test. Different lowercase letters (a, b) represent significant differences between groups (*p* < 0.05).

**Figure 7 nutrients-15-02228-f007:**
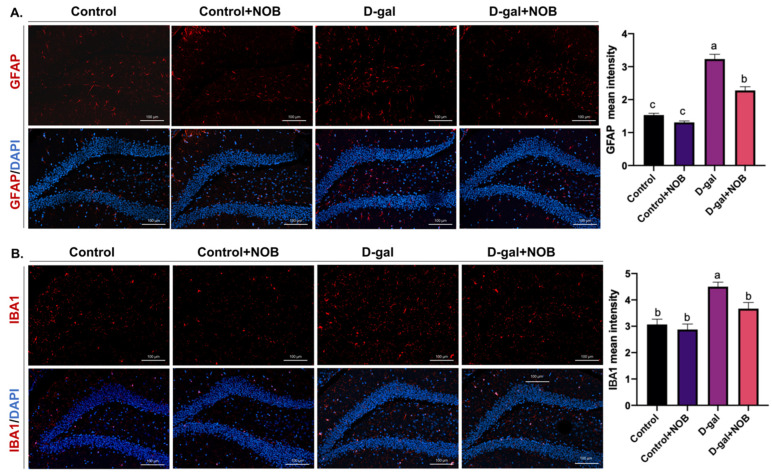
NOB decreased the number of activated microglial and astrocytes in D-galactose-treated mice. (**A**) Representative images of GFAP-positive astrocyte in the DG of the hippocampus and the corresponding quantification graph. (**B**) Representative images of Iba1-positive microglial cells in the dentate gyrus of the hippocampus and the corresponding quantification graph. Values are shown as the mean ± SEM (*n* = 4), from a one-way ANOVA with Tukey’s post hoc test. Different lowercase letters (a, b, c) represent significant differences between groups (*p* < 0.05).

**Figure 8 nutrients-15-02228-f008:**
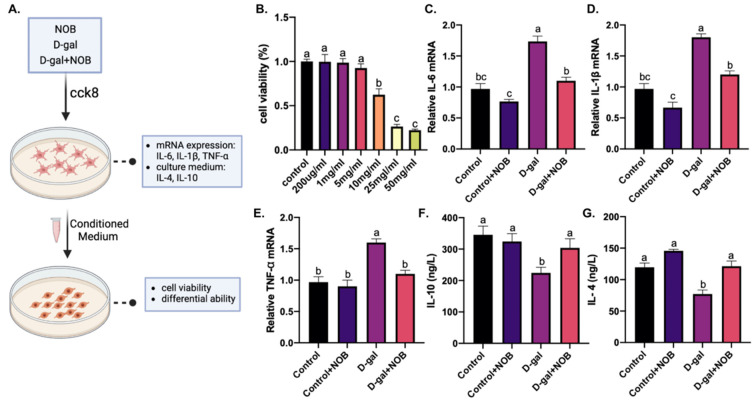
NOB inhibited D-galactose-induced inflammatory responses in BV2 cells (**A**) Schematic illustration of experimental design. (**B**) D-galactose induced BV2 cell death. (**C**–**E**) Relative mRNA levels of IL-6, IL-1β, and TNF-α in the BV2 cells. (**F**,**G**) The levels of cytokines in the supernatant medium of BV2 cells. Values are shown as the mean ± SEM (*n* = 4), from a one-way ANOVA with Tukey’s post hoc test. Different lowercase letters (a, b, c) represent significant differences between groups (*p* < 0.05).

**Figure 9 nutrients-15-02228-f009:**
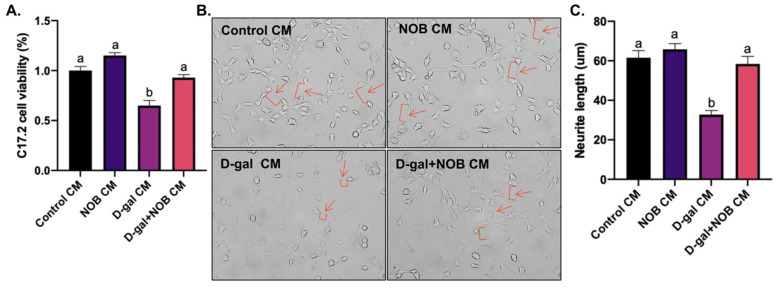
NOB improved D-galactose-induced CM-treated C17.2 cell viability and differential ability (**A**) C17.2 cell viability after treatment with BV2-derived conditioned medium. (**B**) C17.2 cell differentiation in different groups. Red arrows mark the neurites. (**C**) Neurite length was quantified by the tracing method. Values are shown as the mean ± SEM (*n* = 4), from a one-way ANOVA with Tukey’s post hoc test. Different lowercase letters (a, b) represent significant differences between groups (*p* < 0.05).

## Data Availability

The data used during the current study are available from the corresponding author.

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
