# Peer review of "Nobiletin Mitigates D-Galactose-Induced Memory Impairment via Improving Hippocampal Neurogenesis in Mice"

_nutrients, 2023, doi:10.3390/nu15092228_

Round 1

Reviewer 1 Report

In the present manuscript Xiong et al. aims to study the efficacy of nobiletin (NOB), a natural flavonoid extracted from the citrus peels, on aging. For that purpose, D-Galactose was administered to the mice to induce memory impairment and the potential neuroprotective effects of NOB were investigated. The authors investigate the efficacy of the treatments from different perspectives including behaviour using different memory tasks (novel object recognition and spontaneous alternation), immunofluorescence to measure hippocampal neurogenesis, and western blotting to measure neuroinflammation. Although the beneficial effects of NOB on memory have already been previously reported, its neuroprotective effects remain unclear.

The results indicate that animals treated with D-gal showed impaired recognition and spatial memory, impaired neurogenesis in the dentate gyrus, decreased number of neural stem cells and increased levels of neuroinflammatory markers, effects that were all reverted by the co-administration with NOB.

Collectively, the manuscript is well presented, however, I would suggest the revision of an English native speaker as some sentences are not very clear.

I have also some questions and suggestions:

·       How long before the test were the D-gal and NOB administered to the animals? And were the D-gal and NOB administered at the same time in the D-Gal+NOB group?

·       How were the brains chosen for both immunofluorescence and western blot analysis? Was half of the sample for each group assigned to one or the other assay?

·       Where were the BV2 cells obtained from? (i.e what brain region, from which animals?

·       What is the tracing method used to quantify neurite length?

·       The statistical description of the different experiment is a bit poor. The author’s have not described the test they have used to analysed each of the experiments. I am just assuming that in the experiment 1 a ttest comparing both familiar and novel object for each group independently was carried out and for all the other experiments ANOVA were performed. If that’s the case, why are the author’s not reporting any F value, post hoc tests and displaying any asterisk in the graphs? Also, what do the letters on top of each bar mean?

·       I suggest increasing the size of the images a little bit as it is very difficult to see. Also, some of the look like too dark.

Minor comments

Abstract

·       I would replace regress by restore

·       I suggest changing “enhancer to retard brain aging” at the very end of the abstract as you are not measuring retarding of aging in this study.

Intro

·       Change organismal by organism (line 37)

·       Change well-studies by well-study (line 51)

·       Change was administered by were administered (line 91)

·       Replace “For treatment with NOB” and “IN the ends of experiment” as it is not grammatically correct (line 92-93)

Discussion

·       Replace declines by decline (line 311)

·       An “is” is missing after NOB (line 311)

·       Rewrite sentence “Within the DG…” as it is not clear (line 354)

·       Can you add a reference supporting the statement “In response to aging-induced…” (lines 364-366)

Reviewer 2 Report

            The manuscript submitted by Xiong et al. describes the beneficial effects of nobiletin (NOB) in hippocampal neurogenesis. The authors’ goal was to clarify the underlying mechanisms of NOB to protect against D-galactose-induced brain aging, in an effort to support the therapeutic use of NOB for brain aging. They find that NOB improves D-galactose-induced memory deficits, accompanied by the rescue of young neuron number, and amelioration of inflammation. Furthermore, they demonstrate NOB inhibits the release of inflammatory mediators by microglia, which in turn improves neuron survival. The experiments are seemingly well-performed and appropriate; however, there is a noted lack of detail in the explanation of the methods, which has implications on the interpretations and confidence in the data. General and specific comments follow below.

General Comments:

·         The data presented is relevant to the field and the manuscript is reasonably well structured, starting with rescue of memory deficits, moving through additional in vivo analysis of neuron survival and inflammation, and ending with in vitro experiments using conditioned media.

·         Cited references are relevant, and just over half (53%) are recent (within the last 5 years). There is not an excessive number of self-citations.

·         Most significantly to this manuscript, the methods need clarification for replication of the studies and interpretation of the results. Detailed comments on the type of clarifications necessary follow below, but without these details, it is difficult to state with certainty whether the manuscript is scientifically sound and the experimental design is appropriate to test the hypothesis. This lack of information would make it challenging to know whether the results would be reproducible.

·         The figures are appropriate in their overall construction, however, image quality for the in vivo immunohistochemistry needs to be improved. Immunostaining, particularly the red channel, is difficult to see, and the higher magnification insets do not help. Further magnification/higher resolution is necessary. Additionally, schematics for both the in vivo and in vitro experiments that clarify the treatment paradigms are necessary. The figure legends lack the necessary information about the statistical analysis (what significance does “a/b/c/d/” represent?, what is graphed – mean? Standard error or standard deviation? How many animals or independent experiments?). Even though some of this information in included in the Statistical Analysis section of the methods, it should be included in the legends as well. The legends also need to include the magnification of the inset and what the scale bar represents as that is too small to read in the images. Without this, it is difficult to ascertain whether the data is interpreted appropriately.  

·         Given the caveats mentioned above, the conclusions appear to be consistent with the evidence and arguments presented. Whether this manuscript contributes to the field’s understanding of NOB as a potential therapeutic hinges on the issues raised above.

·         The statements regarding ethics and data availability are sufficient.

·         Minor English changes are required. 

Specific Comments:

·         The introduction needs to include a more detailed description of the D-galactose model of aging. There is plenty of literature on its effects, mechanisms, etc, and this information is critical in understanding how NOB may function to rescue the memory deficits following D-galactose.

·         In the animal section (2.1) of the Materials and Methods, there are two major omissions regarding animal usage. The first is that there is no justification for the exclusive use of male mice. This is important as the differences between sexes in aging/neurodegeneration generally, and both neurons and microglia specifically, are widely appreciated and is critical for rigor and reproducibility. The second omission is an explanation for group sizes. There is no mention of power analysis or reference to numbers based on prior published studies.

·         In general, a better sense of the treatment paradigm is necessary. Schematics could be useful. In mice, were NOB and D-galactose given concurrently, or was one started before the other? What was the frequency of subcutaneous D-galactose injections? Was the NOB a single gavage? What was the frequency and duration? This has major implications on prevention versus rescue, and timing of rescue, therapeutic potential, and even more so on the interpretations of the data presented here. Please comment on the dosing and how this relates to human consumption and what is found in the literature.

·         Please include more detailed information on tissue harvest, processing, and storage. Specifically, include details on microdissection, brain fixation for IHC, homogenization for protein analysis.

·         In the Immunofluorescence section of the Materials and Methods (2.3), please include detailed information on sectioning (type – cryosectioning? Brain orientation – sagittal vs coronal? Thickness? Cryopresevation?). How many sections per mouse were used? If more than one, how were they spaced from each other and how did they span/represent the brain? This is significant as there is immense heterogeneity in microglia across brain regions. Please provide details on the imaging – epifluorescence or confocal? Equipment? Magnification? How was hippocampus defined? Please provide details on analysis – what software was used? How were cells counted (those with nuclei only? Within a given area?)

·         Vendor information is needed for many reagents.

·         In Section 2.5, once the conditioned media was collected, was it stored before use? If yes, how so?

·         Figure 1 shows decreased exploration time with D-gal treatment. Please discuss whether this is a consequence of overall locomotion changes or stress/anxiety response, and whether it is typical of D-gal treatment. One would think that if mice are forgetting due to D-gal, they would view the object as novel and spend more time, rather than less, exploring the novel object. Notably, NOB rescues the NOR phenotype, but not the baseline time exploring.

·         Figure 6 is not specifically called out in the text of the results, though the results are presented. Please include a reference to the figure in the text. In addition, the authors note they observed D-gal enhanced activation of microglia and astrocytes. This is not apparent at all in the images presented. It cannot be seen that there is an increase in the overall number of either cell type, the area occupied by glia, the intensity of either marker, or their morphology, any of which would indicate activation. Signals for both Iba1 and GFAP are essentially absent in the control conditions, and while expressed at lower levels, they are there and should be visible.

The quality of the English is ok. Some minor revisions for word choice and grammar are needed, but the authors' meaning is clear as is. 

Reviewer 3 Report

The work from Xiong et al. describes the effects of nobiletin on D-gal induced cognitive, inflammatory and neurogenic damage in wt mice. The work is nicely performed and methods are well explained. Results could be presented more clearly and english language requires some editing. Some considerations:

1) Throughout all the work it seems that neurogenesis is the main reason for cognitive decline associated with brain aging and that, furthermore, this neurogenesis decay is dependent only on neuroinflammation. I would ask the authors to take into consideration, especially in the discussion section, other point of views related to cognitive decline in brain aging.

2) Add at least one reference in the introduction or materials and methods section that D-gal treated mice can be used as a model of senescence.

3) In the materials and methods section, when the authors describe the procedure for the Y-maze, they describe a procedure for a spatial reference version of the test and not a spontaneous alternation version. Closing one arm and then opening will lead the animals to spend a higher % of time in that arm, perhaps even increasing the number of times the animal re-enters that specific. That said, which paradigm did the authors use? The work cited by the authors reports two different protocols for the Y-maze, one related to spatial reference (delayed) Y-maze and one for spontaneous alternations, while authors describe a mixed paradigm. Given the existance of a closed arm, a mixed paradigm may have biased mouse preference for the novel arm, compared to others. Please, clarify. In case authors have used a mixed paradigm, please give more data (not necessarily in the manuscript, it can be added as supplementary or in the reviewer response) on animal performance such as: 1) number of total entries; 2) number of entries in the novel arm compared to the other two arms; 3) time spent in the novel arm compared to one of the other arms.

4) Immunofluorescence representative image are of poor quality: for instance, GFAP signal is hard to appreciate.

5) Letters in each histogram referring to statistical analyses are not explained in the figure legend. Furthermore, there is no also explanation in the statistical section of the material and methods.

6) WB analyses: how are data reported? Because band intensities should be normalized to control levels (this will lead the histogram to be at 1 for control and other columns to be expressed as a variation of control).

7) Figure 8: better quality of the reperesentative image could be appreciated.

8) Statistical p is sometimes described as p<0.0001 (lines 298 and 299). Authors should replace also other instances with p<0.001.

9) Name Phos-P65 phospho P65 or pP65.

English language requires some major editing. The manuscript is comprehensible but many expression are written in a wrong form.

Reviewer 4 Report

The authors explore the molecular mechanism of neuroprotection by Nobiletin (polymethoxylated flavonoid from citrus peel).  It is hypothesized that neurodegenerative activities could be a result of the decrease of neurogenesis that occurs with age. Improving neurogenesis in aged brains may provide therapy for cognitive decline.  The authors use cell culture and a mouse model of age, induced with D-galactose injections to study Nobiletin effects on behavioral memory, inflammation, and neurogenesis.

In an induced age mouse model, mice treated with Nobiletin maintained memory of a previous explored object and also displayed spatial memory retention. From the mice brain tissue, cell counting using fluorescence of immature neurons (DCX+) and stem cell maintenance (SOX2+) showed Nobiletin rescued dysfunction. Western blots of tissue showed increase in inflammatory mediators and additional fluorescent cell counting revealed decrease in microglial and astrocyte activation for the Nobiletin treated groups.

The researchers used BV2 cells as a microglial model and C17.2 cells as stem cells. The BV2 cells treated with Nobiletin had less pro-inflammatory markers and when the cell culture media was added to the stem cell model, those that were treated with Nobiletin were able to differentiate.

The authors research is very exciting and promising for the neuroprotective mechanisms of Nobiletin as seen in age-induced mice and cellular microglial and stem cell models, however, there are issues which need to be addressed in order to fully evaluate the manuscript as outlined below.

Point 1: Authors failed to state where they purchased the cells and reagents used in this research. Please detail where cells, mice and all reagents including ELISA were obtained so that one could evaluate and repeat the experiments with confidence.  Good science publications should be written so that they are fully reproducible.

Point 2:  In Figures 2-8, what do the lowercase a, b and/or c above each error bar represent? This detail is not described anywhere in the manuscript.

Point 3:  Although the authors provided quantification plots, the fluorescent images in Figure 3, Figure 4 and Figure 6 are too dark to be evaluated in this manuscript.  Please replace these representative images that are not dark.

Point 4:  In line 281 it says “supernatant”. Please clarify this use of supernatant.

English language appears good for the majority of the manuscript.

Round 2

Reviewer 2 Report

The authors have sufficiently addressed this reviewer's concerns, thank you. 

Author Response

Thank you for your acknowledgment and confirmation!

Reviewer 3 Report

I thank the authors for the revision performed. The manuscript is now far more understandable compared to the previous version. However, a few things should be changed:

1) In the graphical abstract (or is it figure 1? If it will appear in the text then you should add a Figure 1 legend) change the sentence "Prefer to novel object" to "Preference for novel object".

2) Fig.2 (or Schematic 1) should be referred to as Figure 2 and the legend should be closer to the image.

3) The statistical analysis, for those who will read only the manuscript, is still difficult to understand. You should specify the criteria for significant comparison through which letters are applied also in the statistical part of the methods.

4) Fig. 8: the image is the same of the previous version. Did the authors change it? Because it is not so representative of the reported bar graphs.

English is fine.

Reviewer 4 Report

I am satisfied with the author's corrections and revisions.

English language appears good for the majority of the manuscript.